# Study on the Microstructure and Mechanical Properties of a Ti/Mg Alloy Clad Plate Produced by Explosive Welding

Hui Zhao [1,2,*], Chaochao Zhao [1,2], Yang Yang [1,2], Yizhuo Wang [1,2], Liyuan Sheng [3], Yixu Li [1,2], Miao Huo [1,2], Keren Zhang [1,2], Liwei Xing [1,2] and Ge Zhang [1,2]

1   School of Material Science and Engineering, Xi'an Shiyou University, Xi'an 710065, China; chaoaooo@163.com (C.Z.); yy292642630@163.com (Y.Y.); yz.wang@siat.ac.cn (Y.W.); yixuxuli@163.com (Y.L.); huomiao8888@163.com (M.H.); zkrnwpu@163.com (K.Z.); xingliwei2004@126.com (L.X.); 190609@xsyu.edu.cn (G.Z.)
2   Xi'an Key Laboratory of High Performance Oil and Gas Field Materials, Xi'an Shiyou University, Xi'an 710065, China
3   Shenzhen Institute, Peking University, Shenzhen 518057, China; lysheng@yeah.net
*   Correspondence: huier7921@126.com; Tel.: +86-18149032802

**Abstract:** In this paper, the microstructure and properties of a Ti/Mg alloy clad plate manufactured by explosive welding were studied. The bonding interface was inspected by ultrasonic examination (US). The microstructure and the composition of the clad were characterized by OM and SEM. Properties were inspected by tensile test, shearing test, microhardness test and electrochemical corrosion. The results showed that the bonding interface of the clad plate was made up of straight areas and wavy areas. In straight areas, element diffusion occurred across the bonding interface. Additionally, in wavy areas, a melting zone occurred in the Mg alloy layer near to the bonding interface. Lots of light particles embedded on the melting zone. Tensile test results were comparable with the Ti sheet and the ultimate tensile strength of the clad plate demonstrated an 18% increase. The shearing strength of the clad plate was about 68–87 MPa. The microhardness of the clad plate was higher than that of the original sheets from the interface to 300 μm away. At over 300 μm, the microhardness of the clad plate decreased and approached the original sheets. Compared with the straight area, the hardness of the Mg alloy layer in the wavy area close to the interface increased by 12%. Corrosion results showed that the corrosion potential ($E_{corr}$) absolute value of the clad plate increased by 24%, and the corrosion current density ($i_{corr}$) value was 4 orders of magnitude lower, compared with the Mg alloy sheet. It was clear that the corrosion resistance of the clad plate was higher than that of the Mg alloy sheet. Cladding Mg alloy and Ti by explosive welding would improve the industrial applications of magnesium materials.

**Keywords:** titanium; magnesium alloy; explosive welding; clad plate; microstructure; mechanical property; electrochemical corrosion

## 1. Introduction

Magnesium (Mg) and its alloys possess great potential application in Computer, Communication, Consumer electronic, automotive, biomedical, and aerospace fields due to their low density, high specific strength and stiffness, as well as excellent biocompatibility [1,2]. However, their high chemical activity and poor corrosion resistance limit their further applications. Owing to the porous or loose structure and inferior corrosion resistance of the oxide or hydroxide films generated on the surface, Mg and its alloys are susceptible to corrosion in the environment (acidic solutions or neutral solutions). Thus, it is necessary to take steps to improve the corrosion resistance of Mg and its alloys. Titanium (Ti) and its alloys have excellent corrosion resistance, low density, high specific strength, and heat resistance, and they are widely used in the aviation, aerospace and petrochemical industries. Ti and its alloys exhibit excellent resistance to corrosion attack in many aggressive

media and are deserving of close attention as structural material in the design of chemical processing machinery [3,4]. Thus, it is attractive to clad the Mg alloys with a Ti layer to promote their corrosion resistance. Welding methods for dissimilar materials include fusion welding (such as argon-arc welding, plasma welding and laser welding), friction stir welding, ultrasonic spot welding, and clod metal-transfer welding. However, due to differences in density, melting point and thermal expansion, the fabrication of a layered Mg and Ti clad plate by traditional welding methods is still challenging. Explosive welding is noted for its capability of offering high strength bonding and cladding metals that are unweldable or are difficult to weld by other methods. It has been developed as a promising way to clad dissimilar materials [5–13]. Thus, some attempts have been made to clad other materials on Mg alloys by explosive welding. Arisova [5] reported that aluminum was successfully coated on magnesium alloys by explosive welding and investigated the effects of heat treatment on the nature of change in micro-mechanical properties and phase composition of Mg/Al clad plate. Chen [6] joined the 1100 aluminum and AZ31 magnesium alloy using an explosive welding method and analyzed the welding parameters, including explosive thickness and stand off on the bonding surface. Mro'z [7] reported that a round 22.5 mm-diameter and 160 mm-length Mg/Al bars had been produced using the explosive welding method. The shearing stress values of these Mg/Al bars were about 60 MPa. Additionally, the metallographic examination showed that the bond quality in the samples after the explosive welding process was good. No discontinuities or separations in the bonding region were found. Rouzbeh [8] clad the AZ31B magnesium alloys with the AA1050 aluminum through explosive welding and observed that the interface was wavy, and no intermetallic layers formed in the interface. It indicated that an appropriate bonding was formed between aluminum and magnesium in the interface. However, a few references reported to clad titanium and magnesium alloys by explosive welding. Habib [9] attempted to make a three-layer Ti/Mg/Mg plate by using a Mg alloy as an interlayer material to reduce molten zones. The study showed that a desired wavy interface was difficult to form between the titanium and magnesium. Additionally, some molten zones were still formed in the interface. Zhang [10] used explosive welding to clad Ti and Mg alloys with the aid of an Al interlayer and investigated the bonding mechanism of Ti/Al/Mg clad plate. The study indicated that a periodic wavy between Ti and Al plate. Additionally, the bonding interface between Al and Mg demonstrated a similar wavy shape, but some localized melted zones also existed at this interface. However, for tow-layer explosive welding titanium and magnesium alloy, much work, including the microstructure variations near the bonding interface, mechanical properties and corrosion resistance, is still needed for further investigation.

In this work, a Mg alloy (AZ31B) was coated with pure titanium (ASME SA256 Gr1) by the explosive welding method. The interface microstructure was examined by optical microscopy and scanning electron microscopy. The properties, including tensile strength, shearing strength, microhardness and corrosion resistance, were tested to evaluate the weld quality.

## 2. Experimental Procedure

### 2.1. Materials

Pure titanium (ASME SB265 Gr1) in a solution-annealed condition and a magnesium alloy (AZ31B) in hot rolling condition sheets were used for explosive welding. The chemical compositions of the Ti (ASME SB265 Gr1) and Mg alloy (AZ31B) are listed in Table 1. Ti and Mg alloy sheets were prepared with dimensions of 3 mm × 340 mm × 540 mm and 10 mm × 310 mm × 510 mm, respectively.

**Table 1.** The chemical compositions of Ti and Mg alloy sheets (wt. %).

| Materials | Chemical Element | | | | | | | |
|---|---|---|---|---|---|---|---|---|
| Ti | C 0.06 | | H 0.010 | | N 0.015 | O 0.16 | Fe 0.01 | Ti Bal. |
| AZ31B | Al 2.81 | Si 0.03 | Ca 0.02 | Zn 0.91 | Mn 0.282 | Fe 0.002 | Cu 0.00157 | Ni 0.0005 | Mg Bal. |

*2.2. Explosive Welding Process*

Parallel arrangements were employed for the experiment. Figure 1 shows the assemble drawing of explosive welding. Ti and Mg alloy sheets were placed in parallel directions. The Mg alloy sheet was placed on the anvil and the Ti sheet was placed above with a parallel space of about 6 mm in height. The ANFO explosive layer was spread on the Ti sheet surface. Additionally, the explosives were packed in a wooden box with the same dimensions as the Ti sheet. The thickness of the explosive was 24 mm, and the density was 0.73 g/cm$^3$.The velocity was approximately 2200 m/s. An electronic detonator was placed in the long edge of the Ti sheet. By igniting the electronic detonator, the detonator initiated the explosives, and the Ti sheet was accelerated to rapidly shock the Mg alloy plate. Based on previous research [14], the collision point velocity (Vcp) was 2104 m/s, dynamic collision angle (θ) was 16.2°. Thus, the Ti sheet impact velocity (V$_p$) can be calculated by the following Equation (1) with the V$_{cp}$ and θ [15].

$$V_P = 2V_{cp} \sin (\theta/2) \tag{1}$$

The impact velocity of Ti sheet was 599 m/s.

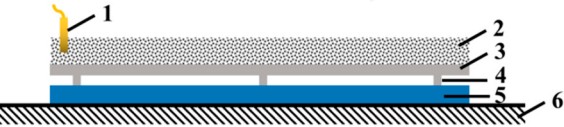

**Figure 1.** A drawing of explosive welding: 1. detonator; 2. explosives; 3. Ti sheet; 4. spacer; 5. Mg alloy sheet; 6. anvil.

*2.3. Specimen Characterization*

After explosive welding, ultrasonic examination (US) was employed to inspect the bonding interface of clad plates with OLYMPUS 5077 equipment [5]. Then, the samples were cut parallel to the detonation direction. The cross-sections of samples were grounded with SiC sandpaper up to No. 2000 and polished with diamond paste. Additionally, they were etched with 3 mL hydrogen nitrate and 97 mL alcohol. The interface microstructure of the clad plate was observed by optical microscopy (Olympus GX51). The interface observation and elements analyses of the clad plate were conducted using scanning electron microscopy (SEM, SSX-550, shimadzu corporation, kyoto, Japan) An energy dispersed X-ray micro-analyzer (EDS) (shimadzu corporation, kyoto, Japan), was also carried out.

Three samples for the shearing test were prepared (shown in Figure 2). Shearing and tensile tests were carried out on a testing machine named DLY-10A, (Shanghai hesheng instrument technology co., ltd., Shanghai, China) The shearing speed was 0.2 mm/min. Microhardness measurement was carried out on the 401MVD Vickers durometer (Shanghai optical instrument factory, Shanghai, China), using a load of 50 g and a dwell time of 15 s. A three-electrode system attached to an electrochemical workstation (CHI660E, Shanghai Chenhua instrument Co., Ltd., Shanghai, China) was used for the electrochemical corrosion measurement. The samples of original Ti sheet, Mg alloy sheet and clad plate were used as the working electrode, while the saturated calomel electrode and platinum electrode were used as reference electrode and counter electrode, respectively. The corrosive medium was 3.5 wt.% NaCl solution at room temperature. The polarization curve was scanned

from $-3.5$ V$_{SCE}$ to 2.0 V$_{SCE}$ at a scan rate of 0.5 mV/s. The corrosion potential ($E_{corr}$) and corrosion current density ($i_{corr}$) for each sample were obtained from the Tafel curve using CorrView software.

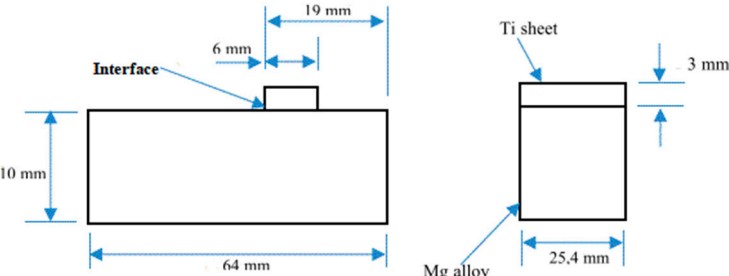

**Figure 2.** The drawing of shearing test sample.

## 3. Results and Discussion

After explosive welding, the Ti/Mg alloy clad plate with 310 mm × 310 mm × 510 mm was obtained. Figure 3 presents the appearance, ultrasonic examination (US) results and optical interface microstructure of the clad plate after explosive welding. Figure 3a shows the appearance of the clad plate. It shows that the Ti and Mg alloy layers are tightly joined together, and no cracks are detected. Ultrasonic examination (US) results also show that the interface reflection wave is obtained, which indicates that the clad plate bonds well (Figure 3b). All the results show that explosive welding is an effective method to join pure Ti and AZ31B Mg alloy sheets. The optical microstructure of the clad plate is shown in Figure 3c. It reveals the interface is made up of wavy areas and straight areas. In straight areas, the interface is smooth and uniform, and no cracks or unbonded defects. In wavy areas, the interface is asymmetric and spikes toward the Ti layer due to the density differences and the inclined collision between Ti and Mg alloy materials [6], but there are still some black areas in the Mg alloy layer close to the interface.

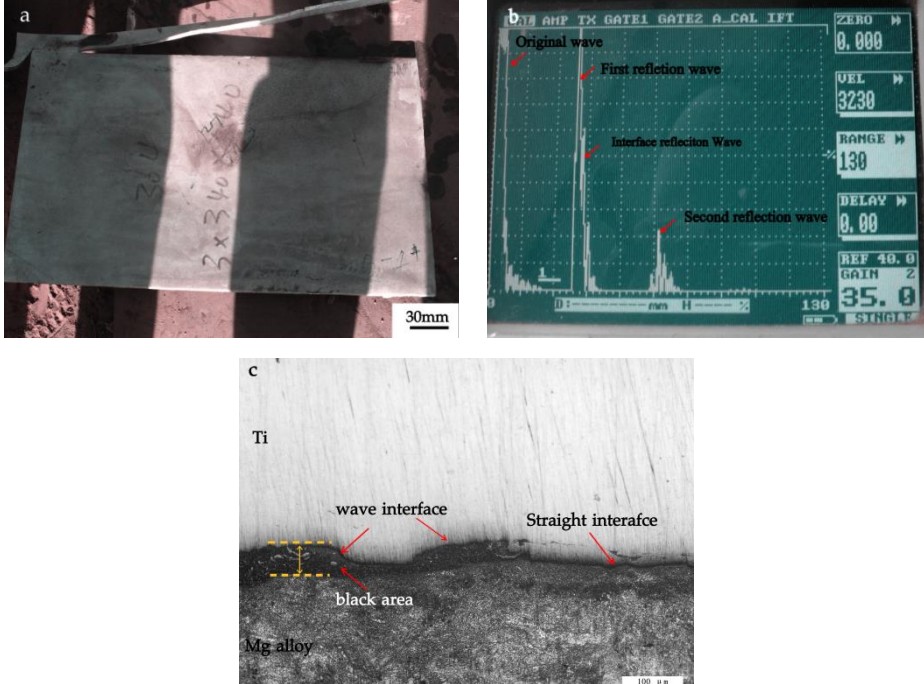

**Figure 3.** The clad plate after explosive welding: (**a**). the appearance; (**b**). the ultrasonic examination (US) results; (**c**). the optical interface microstructure.

To clearly observe the morphology of the bonding interface, SEM and EDS analyses were used (Figure 4). Figure 4a,b shows the straight and wavy areas of the bonding interface, respectively. During the explosive welding process, high temperature and pressure are produced at the collision point, causing the jet to spray in the front of the collision point with a velocity greater than 5000 m/s [6]. Due to inclined collision and resistance of the bonding surfaces, the jet changes its direction to penetrate the bonding surfaces, then the interface forms. Moreover, due to resistance of the bonding interface, the jet separated into two parts: most of the jet moved forward, and the left part penetrated the bonding surface to form the interface. The left part of the jet swirled back in a counterclockwise direction, continually penetrating the bonding surfaces to form straight or wavy interface [16]. The jet penetration is not uniform between participant metals due to the variation in the density and thermal conductivity [17]. The jet for penetration is bigger and the deformation of the bonding surface is more serious. So, the formation of the interface with straight and wavy microstructure is caused by the jet penetration degree. In this paper, both straight and wavy areas are generated in the interface. In straight areas, a straight interface is visible (Figure 4a). Additionally, in higher magnification, as shown in Figure 4c, the interface is uniform, with no cracks or melting zones. However, in wavy areas, the irregular wavy microstructure is demonstrated in Figure 4b. Additionally, there are also no cracks in the interface. However, a melting zone (the black area in Figure 3c) is found on the Mg alloy layer near the interface [10,11]. In higher magnification, Figure 4d shows that lots of light particles embed on the melting zone. Microcracks and pores are also observed in the melting zone. EDS is used to analyze the elemental composition of the melting zone. Figure 4e–g shows the results of EDS analyses for point 1, 2 and 3 in the melting zone. Point 1 is a black area in the melting zone. Additionally, the composition of point 1 is 89.46% Mg, 7.52% O and 3.02% Al (Figure 4e). It is similar to the original Mg alloy sheet. However, the compositions of light particles in the melting zone are different. The shapes of light particles are compact or loose. The EDS analysis of compact particle (point 2) is about 46.22% Mg, 3.56% Al, 36.37% Ti and 13.83% O. Additionally, in the loose particle (point 3), 55.19% Mg, 2.74% Al, 31.78% Ti and 10.29% O are detected. According to the Mg–Ti equilibrium diagram in the literature [18], it is deduced that the possible phases of MgTi and $Mg_2Ti$ are formed in point 2 and point 3.

Element line scanning analyses across the straight and wavy areas in the interfaces were conducted by EDS to investigate the element distribution (Figure 5a,c). The results are shown in Figure 5b,d. Across the straight areas, line scanning analysis indicates that the titanium layer of the clad plate is almost completely Ti and the magnesium alloy layer is almost Mg. However, there are some differences near to the interface. It is noticed that element diffusion occurs across the interface, and no trace of intermetallic compound is observed. During the explosive welding, high temperature (>1000 K) and pressure (>106 Pa) are produced at the collision point, resulting in element distribution [6]. Moreover, there is not enough time to the generate intermetallic compound in the bonding interface due to the fast welding (about $10^{-6}$ s) [6]. Thus, no intermetallic compounds were detected in the straight areas. On the other hand, across the wavy areas, the EDS line scanning results demonstrate that the distribution of Ti in titanium layer and Mg in magnesium alloy layer are similar to those in the straight areas. However, in the melting zone, element distributions are different. During the explosive welding, it is not easy to form intermetallic compounds in the bonding interface. However, many studies show the formation of intermetallic compounds in the melting zone [6,19,20]. Additionally, as mentioned above, the formation of wavy interface is resulted from the high-speed jet (>5000 m/s) penetrating the bonding sheet. In wavy areas, the generated heat is not able to rapidly dissipate, causing the melting zone to form. Thus, the intermetallic compounds occur in this zone, and some micropores and cracks also appear [19]. As shown in Figure 5d, the contents of Ti and Mg in the melting zone are fluctuating along the scanning line. Ti elements only occur in the light particles on the melting zone, and the composition of black area is almost Mg. This is in consistent with the result of EDS point analyses. It has been reported [6,10,21,22] that

the wavy interface is generally preferred in the explosive welding because it provides a larger interface area and higher bonding strength than straight ones. However, for Ti–Mg alloy clad plate, the melting zone, micropores and cracks occur in the wavy interface. They are the weaker locations in the clad plate. Thus, straight interface indicates a good bonding quality for Ti/Mg alloy clad plate.

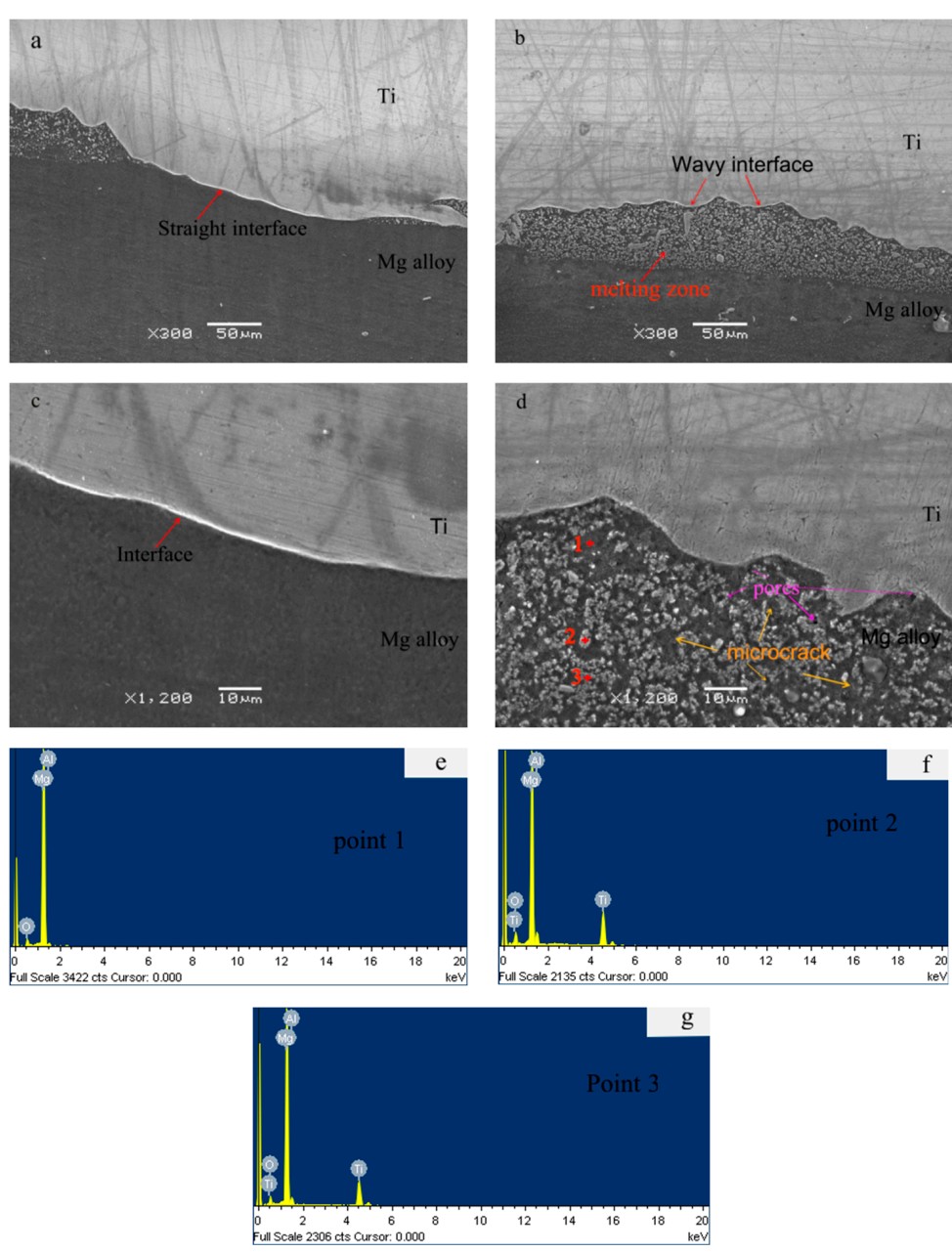

**Figure 4.** Interface microstructure (SEM) and corresponding EDS spectrum analyses: (**a**). straight area; (**b**). wavy area; (**c**,**d**).: enlarged views of (**a**,**b**), respectively; (**e**). EDS analysis of point 1; (**f**). EDS analysis of point 2; (**g**). EDS analysis of point 3.

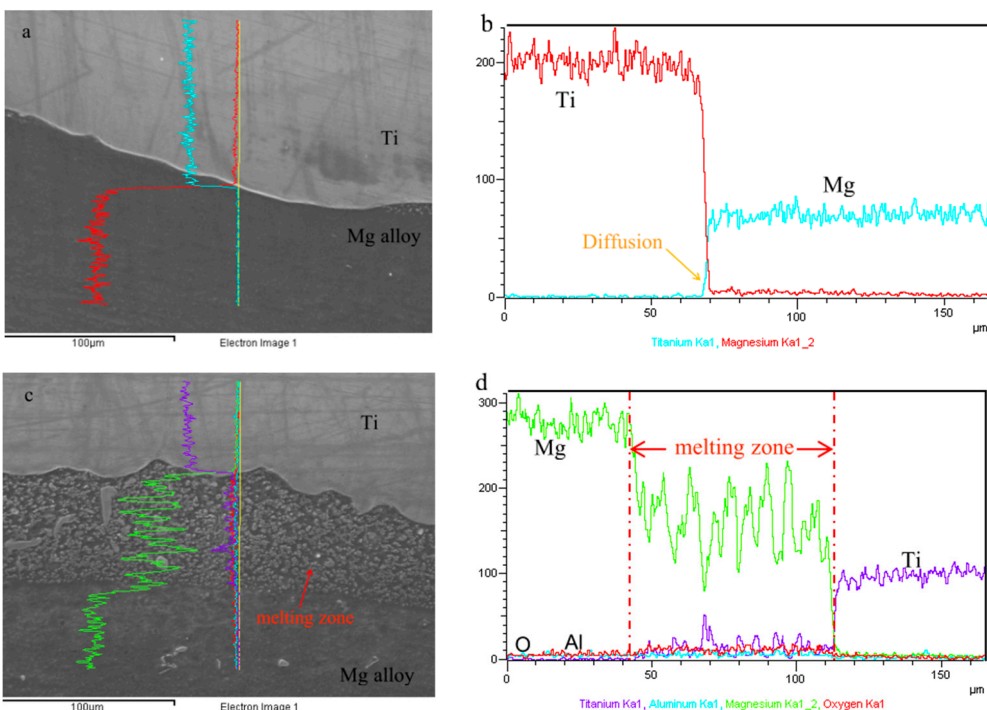

**Figure 5.** Line scanning results: (**a**). straight area of the interface, (**b**). wavy area of the interface, (**c**,**d**). are the line scanning results of straight and wavy areas, respectively.

Figure 6 shows tensile test curves, failed samples and SEM images of fracture surface morphology in the Ti sheet, Mg alloy sheet and clad plate. Stress–strain curves are shown in Figure 6a. It can be seen that explosive welding affects the deformation behavior of the sheets and clad plate. Additionally, the tensile properties of the Ti sheet, Mg alloy sheet and clad plate are summarized in Table 2. An obvious variation occurs in the tensile strength after explosive welding. Compared with the Ti sheet, the ultimate tensile strength increases from 320 MPa to 377 MPa. This shows an 18% increase. Additionally, compared with the Mg alloy sheet, the tensile strength of the clad plate also increases. Above phenomena are mainly due to work hardening from explosive welding. The failed samples of the Ti sheet, Mg alloy sheet and clad plate are shown in Figure 6b. According to the figure, the fractures of the Ti sheet and clad plate demonstrate obvious necking. However, the fracture of the Mg alloy sheet hardly deforms. An important point in this picture is that the separation occurs at the tip of the clad plate fracture, while other areas of the interface continue to join. This indicates that the clad plate performs good bonding quality. The fracture surfaces of the three samples, perpendicular to the tensile loading, are shown in Figure 6c–i. Additionally, as shown in Figure 6c, the fractures of Ti metal demonstrate obvious necking. The fracture surface is composed of a large number of dimples, showing a characteristic nature of ductile mode (Figure 6d), whereas the fracture in the Mg alloy sheet hardly deforms (Figure 6e). In the fracture surface, typical transgranular and cleavage features are witnessed, indicating brittle mode of fracture (Figure 6f). For the clad plate, the shape of the Mg alloy layer changes slightly, and the shape of the Ti layer deforms radically, so there is obvious separation between the Ti and Mg alloy layers (Figure 6g). Additionally, the fractured surface of the clad plate shows a mixed mode consisting of brittle and ductile modes, as reported by Rouzbeh, Z. [8] and Wang, D. [23]. Figure 6h,i show enlarged views of local microstructure in the Ti and Mg alloy layers fractures, respectively. It can be seen that the fracture surface of the Ti layer is composed of a large number of dimples, which confirm the ductile fracture (Figure 6h). Additionally, clearly, at the fracture surface of the Mg alloy layer, fewer shallow dimples are observed. The formation of these shallow dimples indicates a slight plastic deformation in the Mg alloy layer during tensile testing (Figure 6i).

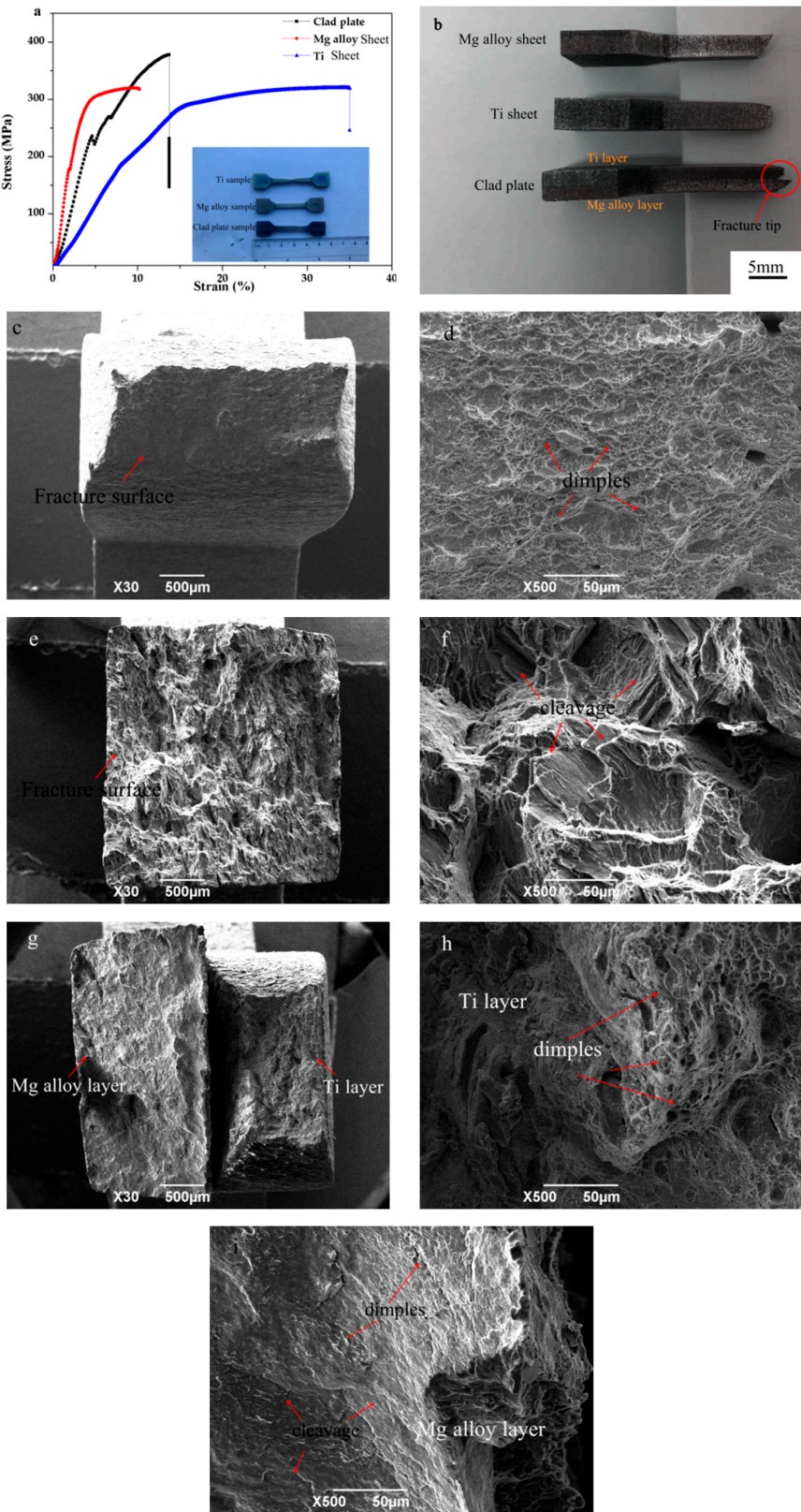

**Figure 6.** Tensile test of Ti, Mg alloy and Ti/Mg clad plates: (**a**). stress–strain curves; (**b**). failed samples after testing; (**c**,**d**): Ti layer fracture surfaces at low and high magnification, respectively; (**e**,**f**): Mg alloy layer fracture surfaces at low and high magnifications, respectively; (**g**). the fracture surface of clad plate at low magnification; (**h**,**i**): the fracture surfaces of Ti layer and Mg alloy layer, respectively.

**Table 2.** The tensile test results of Ti sheet, Mg alloy sheet and the clad plate.

| Sample | Yield Strength (MPa) | Ultimate Tensile Strength (MPa) | Elongation (%) |
|---|---|---|---|
| Ti sheet | 189 | 320 | 34.53 |
| Mg alloy sheet | 174 | 319 | 10.20 |
| Clad plate | / | 377 | / |

Shearing strength is a significant factor for evaluating the bonding interface and welding quality. Three samples for the shearing test are prepared according to Figure 2. Additionally, the measured shear strengths for sample 1, sample 2 and sample 3 are shown in Figure 7. During the shearing strength testing, all failures occur at the bonding interface. The values of sample 1, sample 2 and sample 3 are 68 MPa, 87 MPa and 69 MPa, respectively. Similar results [9,24] have been reported for the shearing strength of the Ti/Mg alloy clad plate.

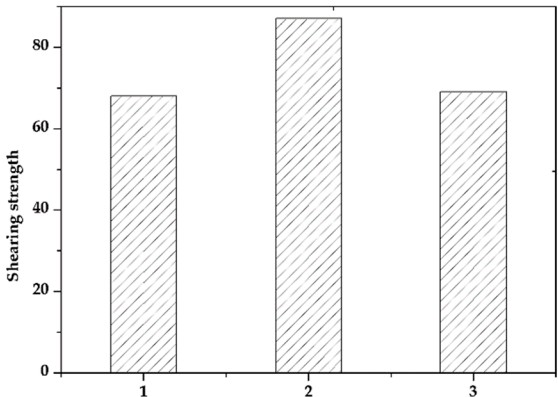

**Figure 7.** Shearing test results of clad plates.

The hardness values of the Ti sheet and Mg alloy sheet are 126 HV and 90 HV, respectively. Hardness measurements are performed across the wavy and straight areas of the interface in the clad plate. Additionally, Figure 8 shows Vickers hardness changes with different distances away from the interface in the clad plate. In straight areas, the hardness value of the Ti layer is 212 HV close to the interface. Towards the thickness center of the titanium layer, the hardness values are 208 HV, 198 HV, 193 HV, 171 HV, 154 HV, 132 HV and 130 HV at 50 μm, 100 μm, 150 μm, 200 μm, 250 μm, 300 μm and 1000 μm away from the bonding interface. In addition, at the same distance in the Mg alloy layer (0 μm, 50 μm, 100 μm, 150 μm, 200 μm, 250 μm, 300 μm and 1000 μm), hardness values are measured, respectively, yielding the values of 117 HV, 109 HV, 101 HV, 97 HV, 99 HV, 93 HV, 90 HV and 91 HV. Most values are higher than original sheets from the interface to 300 μm away. At over 300 μm, the hardness value decreases, and approaches that of the original sheets. Additionally, the maximum microhardness values of Ti and Mg alloy layers all occurs close to the interface. This is due to the deformation resulting from the collision at high velocity during explosive welding [6,17]. In wavy areas, a similar variation of microhardness values is observed. However, compared with the straight areas, the values in the Mg alloy layer are higher from the bonding interface to 100 μm, reaching 131 HV, 128 HV and 106 at 0 μm, 50 μm and 100 μm, respectively. The area is the melting zone in the Mg alloy layer. The hardness value close to the interface even increases by 12%. The increase could be due to the presence of the hard and brittle Ti and Mg intermetallic phases in the melting zone.

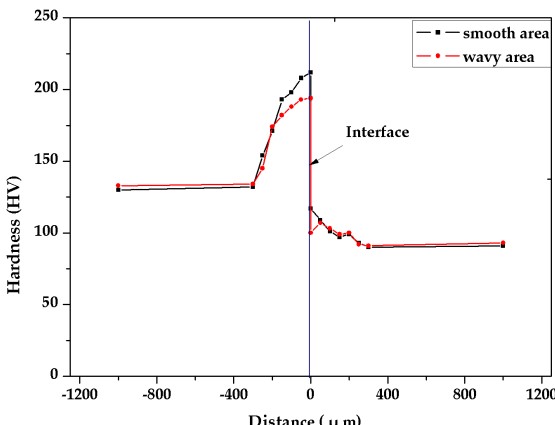

**Figure 8.** Hardness profiles around the interface with wavy and straight structures.

Although it is well known that titanium material has excellent corrosion resistance, a tremendous amount of heat and pressure released by explosives on the surface of the Ti sheet during explosive welding perhaps affects the corrosion resistance of titanium material. For comparison, the corrosion resistance of Ti materials before and after explosive welding is evaluated by electrochemical corrosion test. Figure 9 shows the polarization curves of the original Ti sheet, Mg alloy sheet and clad plate. It can be seen that the Mg alloy sheet possesses active dissolution, and its anodic current density increases rapidly with the increase in anodic overpotential. The original Ti sheet and clad plate behave similarly, having a wide potential range of passive regions. The polarization curves are fitted using CorrView software, and the corrosion potential and corrosion current density are obtained, as shown in Table 3. It shows that the corrosion potential ($E_{corr}$) and corrosion current density ($i_{corr}$) of three samples are different. $E_{corr}$ values of the Ti sheet, Mg alloy sheet and the clad plate are $-746.1$ m$V_{SCE}$, $-1177.7$ m$V_{SCE}$ and $-896.3$ m$V_{SCE}$, respectively. $E_{corr}$ value of the clad plate is more positive than that of the Mg alloy sheet, which increases about 24%. In addition, from the polarization curves parameters (Table 3), $i_{corr}$ values of the Ti sheet, Mg alloy sheet and clad plate are $8.61 \times 10^{-9}$ A/cm$^2$, $1.17 \times 10^{-5}$ A/cm$^2$ and $5.72 \times 10^{-9}$ A/cm$^2$, respectively. The $i_{corr}$ value of the clad plate is 4 orders of magnitude lower than that of Mg alloy sheet. These results indicate the corrosion resistance of the clad plate is much higher than that of the Mg alloy sheet [25,26]. However, some differences are noted between the original Ti sheet and the clad plate. $E_{corr}$ value of the Ti layer in the clad plate is more negative compared with the Ti sheet. The value shows an about 20% decrease. In other words, the corrosion resistance of the Ti material after explosive welding decreases. The phenomenon needs to be investigated in future. Although a reduction in corrosion resistance occurred in the clad plate compared with the original Ti sheet, the clad plate showed a higher corrosion resistance than that of the Mg alloy sheet. Thus, cladding Mg alloy and Ti by explosive welding would help improve the industrial applications of magnesium materials.

**Table 3.** Values of parameters from polarization curves of original Ti sheet, Mg alloy sheet and clad plate (titanium layer as working surface) in 3.5% NaCl solution.

| Materials | $E_{corr}$ (mV) | $i_{corr}$ (A/cm$^2$) |
|:---:|:---:|:---:|
| Ti sheet | $-746.1$ | $8.61 \times 10^{-9}$ |
| Mg alloy sheet | $-1177.7$ | $1.17 \times 10^{-5}$ |
| Clad plate | $-896.3$ | $5.72 \times 10^{-9}$ |

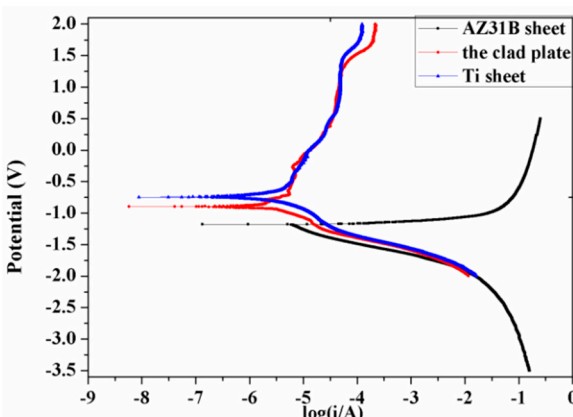

**Figure 9.** Polarization curves of the three samples in 3.5% NaCl solution.

## 4. Conclusions

In this paper, Ti and Mg alloy sheets were joined successfully by explosive welding. Ultrasonic examination (US) results showed that the interface reflection wave was obtained, which indicated that the Ti and Mg layers were tightly joined together. Then, the interface microstructure, mechanical properties and corrosion resistance of clad plate were investigated. The most important results of this paper are summarized as follows:

(1) The interface of the Ti/Mg alloy plate is made up of both straight areas and wavy areas. In straight areas, the element diffusion occurs across the interface. Additionally, in wavy areas, a melting zone occurs in the Mg alloy layer near to the interface. Lots of light particles embed on the melting zone. The compositions of these particles are Mg, Al, Ti and O elements. Based on the EDS analysis results, the possible phases of MgTi and $Mg_2Ti$ are formed in particles.

(2) Tensile test shows that the ultimate tensile strength of the clad plate is 377 MPa. Compared with the Ti sheet, the clad plate shows an 18% increase. The shearing strength values of the clad plate are about 68–87 MPa. The microhardness values of clad plate are higher than original sheets from the interface to 300 μm away. At over 300 μm, the hardness value decreases, and approaches that of the original sheets. Compared with the straight areas in the interface, the hardness value of the Mg alloy layer in the wavy areas close to the interface increases by 12%.

(3) Corrosion results show that the $E_{corr}$ absolute value of the clad plate increases about 24%, and the $i_{corr}$ value is 4 orders of magnitude lower, compared with the Mg alloy sheet. This indicates that the corrosion resistance of the clad plate is better than the Mg alloy. Cladding Mg alloys and Ti by explosive welding would improve the industrial applications of magnesium materials.

**Author Contributions:** Conceptualization, H.Z. and L.S.; methodology, C.Z. and L.X.; software, K.Z.; validation, Y.Y. and Y.W.; formal analysis, H.Z. and C.Z.; investigation, H.Z., Y.L. and M.H.; resources, G.Z.; data curation, C.Z. and Y.Y.; writing—original draft preparation, Y.W.; writing—review and editing, H.Z. and L.S.; visualization, C.Z.; supervision, H.Z.; project administration, H.Z.; funding acquisition, H.Z. and M.H. All authors have read and agreed to the published version of the manuscript.

**Funding:** Natural Science Basic Research Program of Shaanxi (Program No. 2021JM-403, 2021JQ-604); Xi'an Science and Technology Plan Project (Program No. 2020KJRC0100), Scientific Research Program Funded by Shaanxi Provincial Education Department (Program No. 21JC027); Xi'an Shiyou University College Students' Innovation and Entrepreneurship Training Project (Program No. S202010705122, 202110705032), National Natural Science Foundation of China: 5210011310.

**Institutional Review Board Statement:** Not applicable.

**Informed Consent Statement:** Not applicable.

**Data Availability Statement:** Some or all data, models, or code that support the findings of this study are available from the corresponding author upon reasonable request.

**Acknowledgments:** All authors acknowledge the support of Funding.

**Conflicts of Interest:** The author declares there are no conflict of interest regarding the publication of this paper.

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
