# Peer review of "Study on the Microstructure and Mechanical Properties of a Ti/Mg Alloy Clad Plate Produced by Explosive Welding"

_metals, doi:10.3390/met12030399_

Round 1

Reviewer 1 Report

The manuscript "Study on the Microstructure and Mechanical Properties of Ti/Mg alloy clad plate produced by explosive welding" may be of interest to potential readers, but it needs a better presentation of the data and especially of the objective of this work.
Even in the titles there are some spelling errors, which indicates a lack of attention of the authors in the presentation of their work. 

I suggest that the authors withdraw the manuscript and rewrite it in a clearer way, and especially in a more readable one.

Author Response

Thank you for your advises. We paid attention to check the errors in the paper, and corrected errors carefully.

Reviewer 2 Report

- Abstract – “The examination results showed the interface of the Ti-Mg alloy plate was composed of straight 15 and wavy areas. At straight area, the element diffusion occurred across the interface”

Please rearrange the above sentence.

- In general, attention must be paid to language, grammatical errors – poorly written

- line 34 – it is “well” not “weel”

- line 84 – evaluate not evaluated

- fig 6 – please mention samples composition – here it is mentioned as 1, 2 and 3.

- line 258, 257 – why shear strength is averaged if it represents 3 different samples

- line 334 – it is “based” not base

- Line 339 – if there is 60% decrease in elongation, what information authors attempt to convey

- There is a reduction in corrosion resistance compared with original Ti sheet– authors should explain regarding its industrial applications.

Author Response

Thank you for your advises. I have checked carefully the errors in the paper and revised the paper completely as follow:

- Abstract – “The examination results showed the interface of the Ti-Mg alloy plate was composed of straight 15 and wavy areas. At straight area, the element diffusion occurred across the interface”.Please rearrange the above sentence.

Answer:  I have rearranged the sentences in the “- Abstract- ”: lines 15-17

- In general, attention must be paid to language, grammatical errors – poorly written

Answer: I pay attention to check the errors in the paper, and have corrected the paper carefully.

- line 34 – it is “well” not “weel”

Answer: I changed the “weel” to “well”: line 36;

- line 84 – evaluate not evaluated

Answer: I changed the “evaluated” to “evaluate” : line 87

- fig 6 – please mention samples composition – here it is mentioned as 1, 2 and 3.

Answer:   I supplied the drawing of shearing in lines 137-138. And I have mentioned samples number as 1, 2 and 3 in the paper: lines 263-265

- line 258, 257 – why shear strength is averaged if it represents 3 different samples

Answer: I deleted the average value in the paper.

- line 334 – it is “based” not base

Answer: I changed “ base” to “based”: line 350

- Line 339 – if there is 60% decrease in elongation, what information authors attempt to convey

Answer: An reviewer suggested to remove the yield and elongation values, because he think it is not possible to talk about yield and elongation for the clad plate due to the clad is composed. I read the relative engineering specification ( ASTM B898: Standard Specification for Reactive and Refractory Metal Clad Plate), which covers plate consisting of a base metal to which is bonded, integrally and continuously, on one or both sides a layer of one of the following: titanium, zirconium, tantalum, niobium, and their alloys. And some references also supplied only the ultimate tensile strength for the clad plate (kaya Y., Eser G. Production of ship steel-titanium bimetallic composites through explosive welding. Welding in the world, 2019,63(6):1547-1560; Wilson Dhileep Kumar, C., Saravanan S., and Raghukandan K.. Influence of Grooved Base Plate on Microstructure and Mechanical Strength of Aluminum–Stainless Steel Explosive Cladding. Transactions of the Indian Institute of Metals, 2019, 72(12):3269-3276.) So I accept the reviewer’s suggestion and remove the yield and elongation values (lines 237-239). We would like to convey: the tensile strength increase is mainly due to work hardening from explosive welding.

- There is a reduction in corrosion resistance compared with original Ti sheet– authors should explain regarding its industrial applications.

Answer:In this paper, we would like to look for a method to improve the corrosion resistance of the magnesium materials, and improve industrial applications of magnesium materials.The Ecorr absolute value of the clad plate increased about about 24 %, and the icorr value was 4 order of magnitude lower, compared with the Mg alloy sheet. Though a reduction in corrosion resistance occurred in clad plate compared with original Ti sheet, the clad plate showed a better corrosion resistance than that of Mg alloy sheet. So, cladding Mg alloy and Ti by explosive welding, would help improve the industrial applications of magnesium materials. I added it in the paper (lines 330-334)

Reviewer 3 Report

publishable

Author Response

Thank you for your advices.

Reviewer 4 Report

The article is very interesting and presents a different subject or, to a certain extent, not very usual, which is explosion welding. The English and the language used are good. I think the article should be published; however, I list below some points that need to be observed or corrected. Because of it I'm considering a minor review.

Page 1

  • Line 10 – The sentence is confused with a “)” and without a verb. Needs review.
  • Line 18 – Revise this sentence based on comments ahead: “Tensile test results showed that compared with Ti sheet, ultimate tensile strength and yield strength demonstrated the 18% and 17% increases, respectively, and the elongation showed a 60% decrease in elongation.”
  • Line 24 – Replace this sentence “Corrosion results showed that the Ecorr absolute value of the clad plate increased about 24%, and the icorr value was 4 order of magnitude lower, compared with Mg alloy sheet.” By “Corrosion results showed that the corrosion potential (Ecorr) absolute value of the clad plate increased about 24%, and the corrosion current density (icor) value was 4 orders of magnitude lower, compared with Mg alloy sheet.”

Page 2

  • Line 55 replace “…welding. And he investigated the effects…” by “…welding and investigated the effects…”

Page 3

  • Line 103 – remove “our”
  • Line 104 – replace “…was 16.2°. And the Ti..." by "...was 16.2° and the Ti..."
  • Line 112 – Replace “…ultrasonic examination (UT)…” by “…ultrasound examination (US)…”. Do this along all text.

Page 4

  • Line 121 - Explain with a drawing how the shearing test specimens were prepared.
  • Line 149 - Improve image. Avoid shadows.
  • Line 150 – Complementary image (c) with identification of thicknesses.

Page 5

  • Line 154 – Replace “pint” by “point”. Remove “…of the collision point…”
  • Line 169 – Provide reference or explanation at this point of why this “black area” is the “melting zone”. Use material from the text in front.

Page 6

  • Line 211 – Replace “dark area” with “black area”.
  • Line 212 – Remove “indicate”

Page 10

  • Table 2 - It is not possible to talk about “yield” and “elongation” for the clad plate because the material is composed. Two materials been loaded at the same time. It is possible however consider the “ultimate” and relate it´s increase to the effect of the weld, perhaps due to the increase in hardness at the interface. I suggest removing yield and elongation values ​​for clad plate.

Page 11

  • Line 315 – replace “better” by “higher”
  • Line 318 – check if the effect of the decrease in corrosion resistance can be explained by the hardening in the interface.

Page 12

  • Line 337 – considering previous comment remove “221 MPa and 13.66%”

Author Response

Thank you for your advises. I have checked carefully the errors in the paper and revised the paper completely as follow:

Line 10 – The sentence is confused with a “)” and without a verb. Needs review.

Answer: I rearranged the sentences in line 10: In this study, an investigationwas carried out to characterize the microstructure and properties of Ti-Mg alloyclad plate by using explosive welding.

  • Line 18 – Revise this sentence based on comments ahead: “Tensile test results showed that compared with Ti sheet, ultimate tensile strength and yield strength demonstrated the 18% and 17% increases, respectively, and the elongation showed a 60% decrease in elongation.”

Answer: I accepted your suggestion and remove the yield and elongation of the clad plate. Thus I rearranged the sentences in lines 18-20: Tensile test results showed that, compared with Ti sheet, ultimate tensile strength of the clad plate demonstrated a 18% 

  • Line 24 – Replace this sentence “Corrosion results showed that the Ecorr absolute value of the clad plate increased about 24%, and the icorr value was 4 order of magnitude lower, compared with Mg alloy sheet.” By “Corrosion results showed that the corrosion potential (Ecorr) absolute value of the clad plate increased about 24%, and the corrosion current density (icor) value was 4 orders of magnitude lower, compared with Mg alloy sheet.”

 Answer: I rearranged the sentences in lines 25-27: Corrosion results showed that the corrosion potential (Ecorr) absolute value of the clad plate increased about about 24%, and the corrosion current density (icorr) value was 4 order of magnitude lower, compared with Mg alloy sheet.

Page 2

Line 55 replace “…welding. And he investigated the effects…” by “…welding and investigated the effects…”

 Answer: I rearranged the sentences in lines 56-59: Arisova [5] reported that aluminum AD1 was successfully coated on the magnesium alloy MA2-1 by explosive welding and investigated the effects of heat treatment on the nature of change in micro-mechanical properties and phase composition of Mg-Al clad plate.

Page 3

Line 103 – remove “our”

Line 104 – replace “…was 16.2°. And the Ti..." by "...was 16.2° and the Ti..."

Line 112 – Replace “…ultrasonic examination (UT)…” by “…ultrasound examination (US)…”. Do this along all text.

Answer: I removed the “our” in line 106.

Answer: I replaced “…was 16.2°. And the Ti..." by "...was 16.2° and the Ti...": lines 107-108

Answer: I replaced“…ultrasonic examination (UT)…” by “…ultrasound examination (US)…”. And I did this along all text: in lines 12, 116, 141,144, 156 and 339.

Page 4

Line 121 - Explain with a drawing how the shearing test specimens were prepared.

Line 149 - Improve image. Avoid shadows.

Line 150 – Complementary image (c) with identification of thicknesses.

Answer: I supplied the drawing for the shearing test sample: Figure 2 in lines 137-138.

Answer: I took the picture in the work spot. Now the clad plate was cut to make samples. In future study, I would avoid shadow when I take picture.

Answer:  I supplied the identification of thicknesses.

Page 5

Line 154 – Replace “pint” by “point”. Remove “…of the collision point…”

Line 169 – Provide reference or explanation at this point of why this “black area” is the “melting zone”. Use material from the text in front.

Answer: I changed the “pint” by “point” in line 162.

Answer: I provided the references in line 178.

Page 6

Line 211 – Replace “dark area” with “black area”.

Line 212 – Remove “indicate”

Answer: I replaced “dark area” by “black area” in line 220.

Answer: I remove “indicate” in line 221.

Page 10

Table 2 - It is not possible to talk about “yield” and “elongation” for the clad plate because the material is composed. Two materials been loaded at the same time. It is possible however consider the “ultimate” and relate it´s increase to the effect of the weld, perhaps due to the increase in hardness at the interface. I suggest removing yield and elongation values ​​for clad plate.

Answer: I removed the yield and elongation value for clad plate: lines 236-239.

Page 11

Line 315 – replace “better” by “higher”

Line 318 – check if the effect of the decrease in corrosion resistance can be explained by the hardening in the interface.

Answer: I replaced “better” by “higher” in line 331

Answer: It is a good advise to check the hardening effect on corrosion resistance. I would take some supplementary tests to check the effect for further study in the future.

Page 12

Line 337 – considering previous comment remove “221 MPa and 13.66%”

Answer: I removed the yield and elongation value for clad plate: lines 351-351.

Round 2

Reviewer 1 Report

The authors made a heavy revision of the manuscript, but there are still some parte to be improved.

Title : Please correct  "...on the Microstruceture.." and also it seems that the title contains a link to "https://link.springer.com/article/10.1007/s12666-021-02433-0". I don't know why (!!!)

Line 156. Please , do not start the sentence  "but". I suggest " ...between Ti and Mg alloy materials [6],  but there are still some black.."

Line 348-349. Please improve the sentence. I suggest "Ultrasonic examination (US) results showed that the interface reflection wave was obtained, which indicated that Ti and Mg.."

line 353 . I suggest " alloy plate is made up of both straight and wavy areas.."

Figure 8. Please improve the text quality  for  axes and legend.

Author Response

Thank you for your advises. I have checked carefully and revised the paper completely as follow:

Title : Please correct  "...on the Microstruceture.." and also it seems that the title contains a link to "https://link.springer.com/article/10.1007/s12666-021-02433-0". I don't know why (!!!)

Answer: I check the title and corrected it.

Line 156. Please , do not start the sentence  "but". I suggest " ...between Ti and Mg alloy materials [6],  but there are still some black.."

Answer: I revised the sentence:“ And in wavy area, the interface is asymmetric and spikes toward Ti layer due to the density differences and the inclined collision between Ti and Mg alloy materials [6], but there are still some black areas in Mg alloy layer close to the interface.” (lines 152-153)

Line 348-349. Please improve the sentence. I suggest "Ultrasonic examination (US) results showed that the interface reflection wave was obtained, which indicated that Ti and Mg."

Answer: I rearrange the sentence:"Ultrasonic examination (US) results showed that the interface reflection wave was obtained, which indicated that Ti and Mg."(lines 341-342)

line 353 . I suggest " alloy plate is made up of both straight and wavy areas.."

Answer: I rearrange the sentence:"alloy plate is made up of both straight and wavy areas."

(line 346)

Figure 8. Please improve the text quality  for  axes and legend.

Answer: I have improved the text quality for axes and legend for Figure 8 (line 305).